# The smfBox is an open-source platform for single-molecule FRET

Benjamin Ambrose [1,4], James M. Baxter [1,4], John Cully [1,4], Matthew Willmott [1], Elliot M. Steele [2], Benji C. Bateman [3], Marisa L. Martin-Fernandez [3], Ashley Cadby [2], Jonathan Shewring[1], Marleen Aaldering[1] & Timothy D. Craggs [1✉]

Single-molecule Förster Resonance Energy Transfer (smFRET) is a powerful technique capable of resolving both relative and absolute distances within and between structurally dynamic biomolecules. High instrument costs, and a lack of open-source hardware and acquisition software have limited smFRET's broad application by non-specialists. Here, we present the smfBox, a cost-effective confocal smFRET platform, providing detailed build instructions, open-source acquisition software, and full validation, thereby democratising smFRET for the wider scientific community.

[1] Sheffield Institute for Nucleic Acids, Department of Chemistry, University of Sheffield, Sheffield, UK. [2] Department of Physics, University of Sheffield, Sheffield, UK. [3] Central Laser Facility, Research Complex at Harwell, Rutherford Appleton Laboratory, Oxford, UK. [4] These authors contributed equally: Benjamin Ambrose, James M. Baxter, John Cully. ✉email: t.craggs@sheffield.ac.uk

FRET is a photophysical process which results in the transfer of excitation energy from a donor fluorophore to an acceptor chromophore[1]. The efficiency of this transfer process scales inversely with the sixth power of the distance between the two chromophores. Therefore, by measuring the FRET efficiency (e.g. by observing the emission of the two fluorophores under excitation of the donor), spatial information can be determined in the 3–10 nm range, making FRET a 'spectroscopic ruler'[2] well matched to the dimensions of biomolecules such as nucleic acids and proteins[3]. In ensemble measurements this can be used to detect on-off/relative distance changes such as binding and cleaving in bimolecular interactions, or conformational changes (e.g. opening and closing) in unimolecular processes. At the single-molecule level FRET is sensitive to heterogeneous subpopulations, can measure kinetics of processes at equilibrium[4], and as demonstrated by a recent inter-laboratory benchmarking study[5], absolute FRET efficiencies can be used to infer precise distances for biomolecular structure determination[6–11].

Two experimental formats are commonly employed to obtain smFRET data: a confocal approach, in which individual molecules are detected as they diffuse through a confocal volume[3]; and a total internal reflection fluorescence (TIRF) microscopy approach[12], in which individual molecules are immobilised on a glass coverslip and excited by an evanescence field. The two approaches are largely complementary, together allowing the interrogation of biomolecular dynamics at timescales spanning twelve orders of magnitude: from the picosecond–millisecond (confocal); and millisecond–hours (TIRF) (see ref. [13] for a recent review). Confocal experiments are often the first smFRET experiments performed on a new biomolecular system, as this is the more straightforward approach, not requiring surface immobilisation (which can be non-trivial). Such experiments can reveal different biomolecular conformations present in (dynamic) equilibrium, and the corresponding rates of conformational transitions, at timescales relevant to key cellular processes such as protein (un)folding[14], transcription[15] and DNA replication and repair[16,17]. Confocal experiments are also the approach of choice for generating multiple smFRET restraints for integrative structural modelling[6–11], due to the simpler sample preparation, fast data acquisition, and higher time resolution.

Despite the many advantages of smFRET, it is currently rarely used outside specialist labs, largely due to the high costs of commercial instruments and lack of self-build, easy to use alternatives. To address this, here we provide detailed build instructions, parts lists, and open-source acquisition software, to enable a broad range of scientists to perform confocal smFRET experiments, on a validated, self-built, robust and economic instrument.

## Results and discussion

Here we present the *smfBox*[18], a cost-effective confocal-based platform capable of measuring the FRET efficiency between dye pairs on freely diffusing single molecules, using variable alternating laser excitation (ALEX)[19] for verification of correct dye stoichiometry and the determination of accurate FRET correction factors[5,20,21]. The smfBox (Fig. 1a, b) is constructed from readily-available optics and optomechanical components, replacing an expensive microscope body with machined anodised-aluminium, which forms a light-tight box housing the excitation dichroic, objective, lenses, and pinhole (see Supplementary Note 1, Supplementary Movie 1 and online[18] for the complete parts list, and animated building and alignment protocols). When assembled, the smfBox is sufficiently light-tight to allow safe and effective operation under ambient light conditions, as a Class I laser product (eliminating the need for user laser-safety training). The smfBox can be operated with either customisable LabVIEW acquisition

software, or a stand-alone user interface written in C++. Both versions of the software provide all the necessary functionality for setting up the microscope (alignment and focusing) and recording data (Fig. 1c and Supplementary Notes 2 and 3). The raw data, comprising photon arrival times and detector ID, are saved in the open-source photon-HDF5 data format (Supplementary Note 4)[22], and can subsequently be analysed either with the FRETBursts python module[23] using the Jupyter Notebooks provided (Supplementary Note 6 and online[18]), or with the GUI-based MATLAB package PAM[24].

The smfBox detects both donor and acceptor emission from single molecules freely diffusing through a confocal spot under alternating laser excitation (Fig. 1d, e). Emission under green excitation is used to determine the FRET efficiency (E; Supplementary Equation 1), whilst the response to the red laser confirms the presence of an active acceptor on the molecule, and allows the calculation of the stoichiometry parameter (S; Supplementary Eq. 2) and all correction parameters required for accurate FRET determination (Supplementary Eqs. 3–5)[5,20,21]. The precise ALEX cycle of the smfBox can be fully customised, allowing for faster or slower cycles, periodic acceptor excitation (PAX)[25], or an asymmetric ALEX scheme, which we show can reduce the width of FRET histograms, thereby increasing the resolution of different FRET species (Supplementary Note 7). Furthermore, the design of the smfBox includes a 10:90 beam splitter in the excitation path, directing excitation light to a photodetector (Fig. 1a) to allow for precise monitoring of both laser powers in real time during the experiment, which can be saved into the HDF5 file. Our default optimal values of the laser powers, iris diameter, and ALEX cycle are provided for this setup (see methods).

To test the performance of the smfBox we measured the FRET efficiencies of three DNA standards (Fig. 2a), which were recently characterised by multiple labs using a range of commercial and home-built microscopes[5]. Using the published correction procedures implemented in our open-source python analysis (Jupyter notebooks - Supplementary Note 6), we obtained data in excellent agreement with those from the other labs in the blind study. This provides both an excellent validation of the smfBox, but also a useful diagnostic for users to test their own builds of this instrument, as the successful reproduction of these data means that all hardware, acquisition and analysis software must be working correctly.

Furthermore, we have demonstrated the capability of the smfBox to recover rates of molecular conformational dynamics, using DNA hairpins as a test system (Fig. 2b). DNA hairpins have been shown to interconvert between a closed and open state, with rates that are dependent on NaCl concentration[26,27], within time scales accessible to smFRET experiments (see Supplementary Note 8). First, we reproduced data for a hairpin used in a recent study[26], using dynamic photon distribution analysis (dPDA)[4] to determine opening and closing rates for a series of NaCl concentrations (Fig. 2c). Next, we analysed two further hairpins (Fig. 2d), identical in DNA sequence, but with a smaller and greater inter-fluorophore distance in the closed state, to test the effects of the magnitude of FRET efficiency changes on the precision of the recovered kinetic parameters. As might be expected, the rates of interconversion for the hairpin with a higher-FRET-efficiency closed state could be determined more precisely, whereas analysis of the lower-FRET hairpin produced more variable results (Fig. 2e). In cases where the static (low-FRET and high-FRET) species have considerable overlap with each other (and therefore with the dynamic population) the dPDA model is less well constrained, leading to a greater variation in the values of recovered rates for a given sample size. These results have implications for the optimal positioning of FRET dyes when designing dynamic experiments.

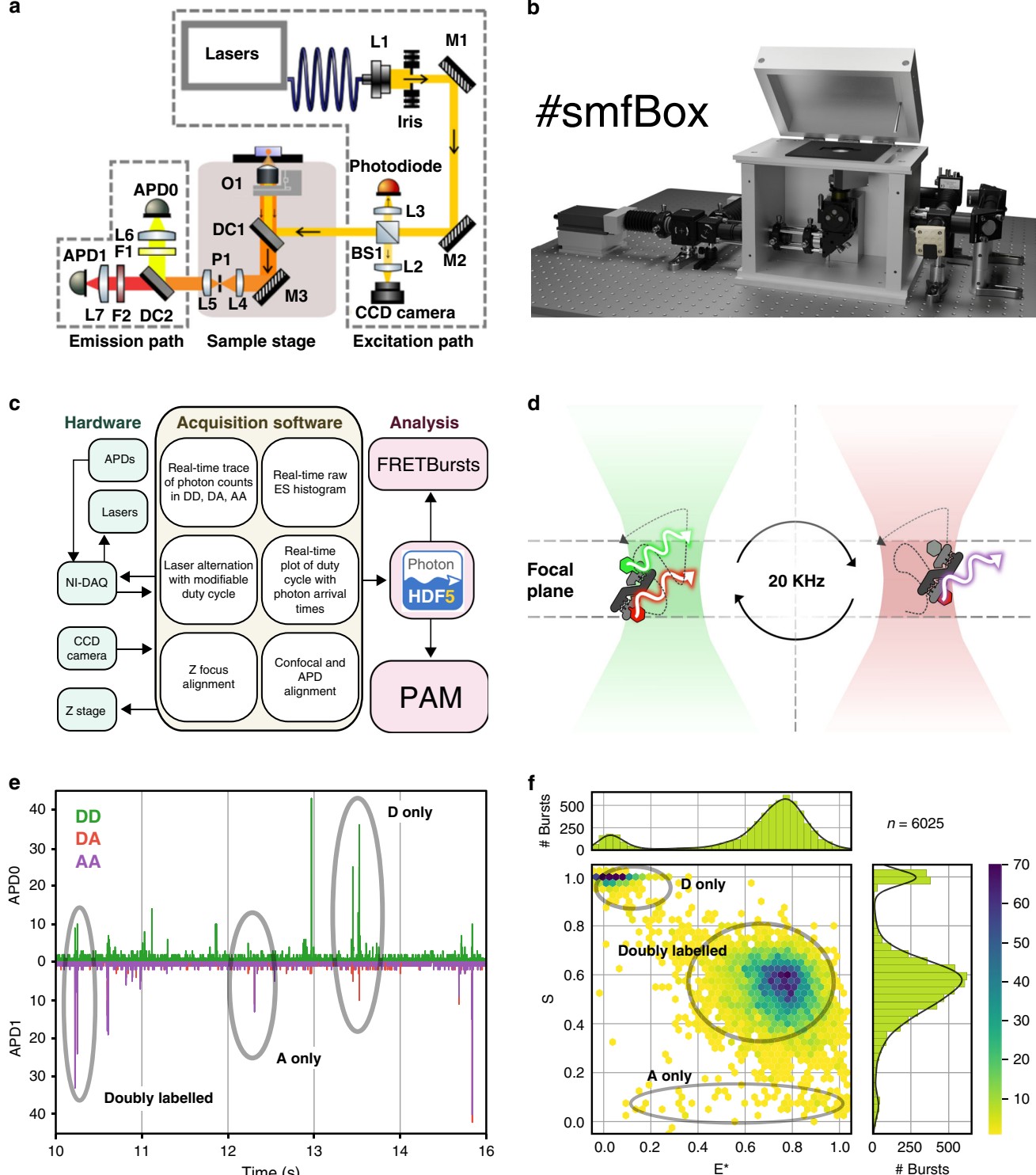

Whilst the microscope we describe is built for ALEX confocal smFRET, its modular design makes for easy expansion to several related techniques. Without any additional hardware, the smfBox is capable of fluorescence correlation spectroscopy (FCS—using a single continuous-wave laser) and fluorescence cross-correlation spectroscopy (FCCS—using two lasers). We demonstrate this capability by determining the diffusion constant for a duplex DNA (Supplementary Note 10). The addition of one or more pulsed lasers and time-correlated single-photon counting (TCSPC) electronics will enable fluorescence lifetime correlation spectroscopy (FLCS) and pulsed-interleaved excitation (PIE) experiments[28]. The further addition of polarisation filters and two additional APDs would constitute a full multi-parameter fluorescence detection (MFD) setup[29]. Furthermore, the addition of an XY-stage to the Z-positioning stage already included would facilitate any number of imaging techniques by scanning the sample. A list of recommended components for such expanded applications can be found in Supplementary Note 9.

In conclusion, we have provided all necessary instructions and software required for the construction and operation of the

**Fig. 1 The smfBox and smFRET. a** Schematic of the smfBox with parts labelled according to Supplementary Tables 1, 2 and 4. Lasers are collimated (L1), cropped (Iris) and steered by two mirrors (M1, M2) onto a 10:90 beam splitter (BS1). 10% of the beam is focused on to a photodiode for continuous power measurement and alternation cycle monitoring. 90% of the beam is directed via dichroic mirror 1 (DC1) to the objective. Light from the back reflection is reflected by DC1 and BS1 onto a CCD camera for accurate focusing. Fluorescence emission from the sample passes through the excitation dichroic (DC1) and is focused onto a pinhole (P1) to remove out of focus light, before being split by colour (DC2) onto one of two avalanche photodiodes (APD0, APD1). **b** 3D model of the completed smfBox, with the front panel of the microscope body removed. A CAD file of the assembled instrument is provided (Supplementary Data 1) along with an animated build sequence (Supplementary Movie 1). **c** A flowchart of the smfBox platform showing the functionality of the acquisition software. **d** Schematic: single molecules diffuse through a confocal volume (residence time ~1 ms) constructed by focusing the lasers into a near-diffraction-limited spot and using a pinhole to section the emission light in a thin focal plane. Lasers (515 nm and 638 nm) are alternated (20 kHz) to ensure multiple excitations of the donor and acceptor dyes for each molecule. **e** A typical time trace for an smFRET experiment. Fluorescently-labelled molecules diffuse through the confocal volume emitting bursts of fluorescence. Individual photons generated by emission from the donor under donor excitation (DD—green) are recorded on APD0. APD1 records photons emitted by the acceptor either under donor excitation (DA—red) or direct acceptor excitation (AA—purple). **f** 2D ES histogram showing uncorrected FRET efficiency (E*) and stoichiometry (S). Donor-only molecules appear with low E* but high S, and acceptor-only molecules appear with low S. Doubly labelled molecules appear with intermediate S.

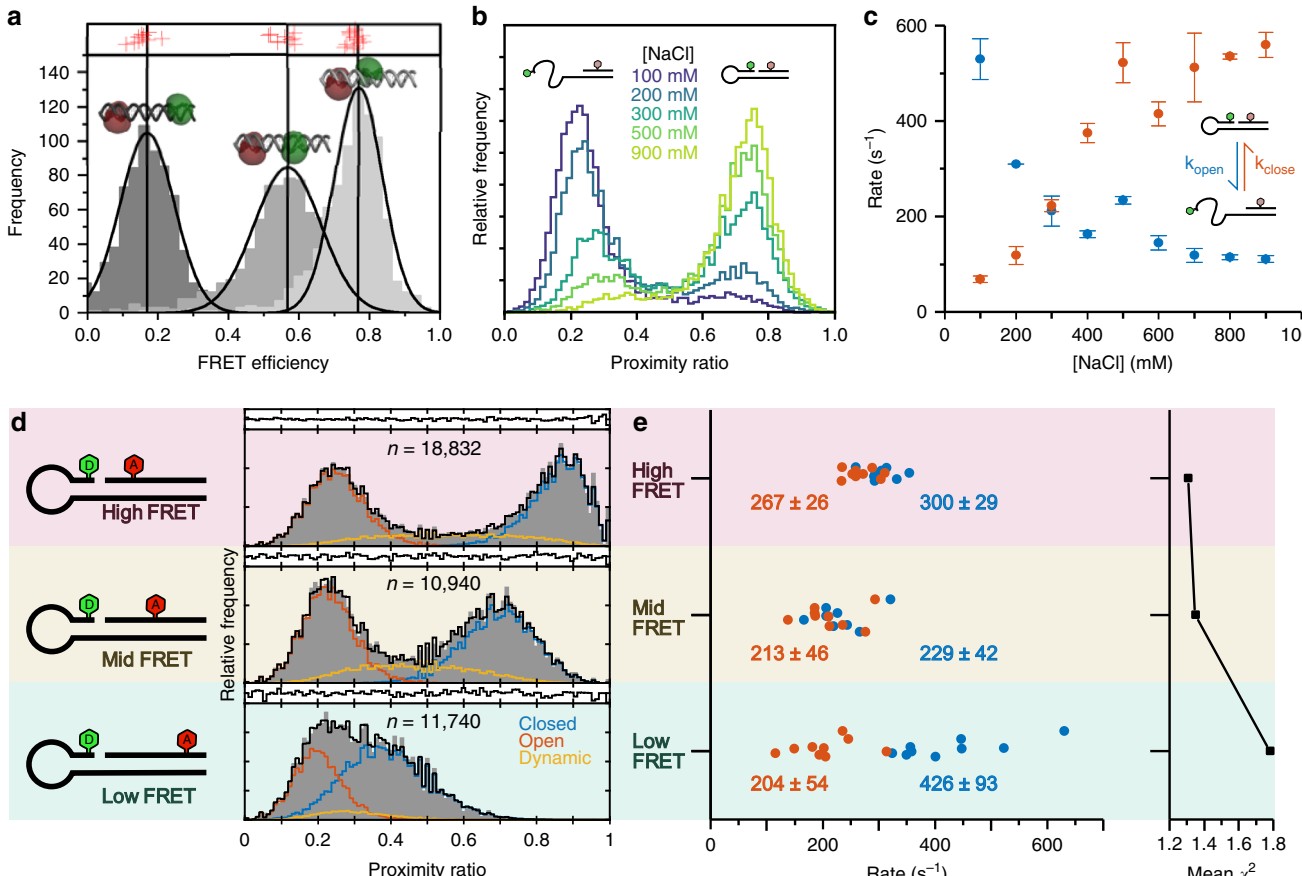

**Fig. 2 Experiments validating the smfBox. a** Fully corrected FRET efficiency histograms of three doubly labelled DNA standards (1a, 1b and 1c, cartoons with dye accessible volumes; for sequences see Supplementary Note 5) measured using the smfBox (grey). Vertical black lines and curves show Gaussian fits of our data, $E = 0.17 \pm 0.07$, $E = 0.57 \pm 0.1$, $E = 0.77 \pm 0.07$ (mean ± sd), compared to the results from 20 other labs as part of a multi-lab benchmarking study[5] (red crosses—Supplementary Note 6). **b** Proximity ratio (uncorrected FRET efficiency) histograms of a DNA hairpin at indicated salt concentrations (see Supplementary Note 5). **c** Salt dependent rates for hairpin opening ($k_{open}$) and closing ($k_{close}$) determined by dynamic photon distribution analysis (dPDA)[4] (mean ± SD, $n = 2$ with >1000 molecules per technical repeat at each [NaCl])—Source data are provided in a Source Data file. **d** Proximity ratio histograms of High-, Mid- and Low-FRET hairpins (at 300 mM NaCl). Data (grey) were fit using dPDA (black) to a two-state model, comprising a closed population (blue), open population (orange) and interconverting dynamic population (yellow). **e** Plot of rates determined from dPDA of nine data sets for each hairpin, each containing 2000 molecules, quoting the mean and standard deviation across the data sets, with the mean chi-squared of the fits plotted to the right.

smfBox, a cost-effective and open-source smFRET microscope with competitive capabilities[18]. We demonstrate the smfBox can determine absolute FRET efficiencies with the same accuracy as other instruments used by the community[5], and can recover biomolecular interconversion kinetics in the range ~50–500 s⁻¹,

in agreement with previous studies[26]. We have shown that an asymmetric ALEX duty cycle can reduce the width of smFRET histograms, increasing the resolution of different FRET species. Finally, we have experimentally assessed the ability to determine kinetic rates of interconversion using dPDA, for systems with

differing magnitudes of FRET efficiency changes, providing useful information for the design of dynamic smFRET experiments. We anticipate that our low-cost open-source approach will ease the adoption of smFRET by the wider scientific community.

## Methods

**The smfBox**. Full details of the construction and operation of the smfBox are described in Supplementary Note 1, Supplementary Figs. 1–14, Supplementary Movie 1 and online[18]. Briefly, the smfBox alternates two lasers (515 nm–222 μW, and 635 nm–68 μW, Omicron LuxX plus lasers, powers measured immediately before the excitation dichroic) by TTL-controlled modulation of electronic shutters. The beams are coupled into a single-mode fibre before being collimated (to 10 mm) and cropped by an iris (to 5 mm), then directed into a custom built anodised-aluminium microscope body (see Supplementary Figures 1–14, and Supplementary Data 1 and 2 for technical drawings and a CAD file of the assembled instrument). A dichroic mirror (Chroma ZT532/640 rpc 3 mm) directs the beam into an objective (Olympus UPLSAPO ×60 NA = 1.35 oil immersion), and the same objective collects the emission, which is focussed onto a 20 μm pinhole and split (Chroma NC395323—T640lpxr) to two avalanche photodiodes (SPCM-AQRH-14 and SPCM-NIR-14, Excelitas), where photon arrival times are recorded by a national instruments card (PCIe-6353).

**Accurate FRET experiments**. Three duplex DNA constructs (referred to as 1a, 1b and 1c) labelled with Atto550 (donor) and Atto647N (acceptor), and were provided by the Hugel Lab as part of the blind, multi-lab FRET study[5] (for sequences see Supplementary Note 5). DNAs were diluted to approximately 100 pM in observation buffer 1 (20 mM $MgCl_2$, 5 mM NaCl, 5 mM Tris, pH 7.5), and ~50 μl placed on a coverslip passivated with 1 mg/ml BSA, and data were acquired by the smfBox. Analysis was done in Anaconda 5.3.0, with Jupyter Notebooks using the FRETBursts python module[23] (version 0.6.5). Background in each channel was estimated by means of an exponential fit of inter-photon delays. Bursts were identified using an all photon sliding window algorithm previously described[23,30] with $L = 10$ and $F = 45$ for both channels, and background was subtracted. Spectral cross talk factors were found by combining data from all standards and extracting bursts with a stoichiometry >0.95 as the donor only population and <0.175 for the acceptor only population (see Fig. 1f) to calculate α and δ, respectively. A dual channel burst search (DCBS) was then used to extract doubly-labelled bursts from each 30-min acquisition, and used to find E and S with single Gaussian fits. Combined data from all three oligos were then plotted together and fitted to obtain γ and β (Supplementary Equation 5). Corrected FRET efficiencies of all doubly-labelled bursts were then obtained using all four correction parameters as previously described[5]. See Supplementary Note 7 for more details. Jupyter Notebooks and raw data (HDF5 files) are available on the smfBox github[18].

**Hairpin dynamics**. DNA hairpins were made from a self-complementary oligo-nucleotide labelled with Cy3B annealed to a short oligonucleotide with Atto647N at different positions (see Supplementary Note 5). Oligonucleotides were purchased from LGC Biosearch (UK), with internal amino modified–dT bases which were labelled with Cy3B and Atto647N NHS esters purchased from GE Healthcare (US) and ATTO-TEC (Germany), respectively. Labelled DNAs were purified via polyacrylamide gel electrophoresis and annealed in annealing buffer (50 mM NaCl, 10 mM Tris HCl, pH 7.5, 1 mM EDTA), by heating to 95 °C (5 min) followed by overnight cooling. Hairpins were then diluted to approximately 100 pM in Hairpin buffer (Tris 50 mM pH 7.5, BSA 0.1 mg/ml, EDTA 1 mM, Glycerol 5%, DTT 1 mM) with additional NaCl where specified, placed into a chamber made of two coverslips and a silicone gasket (to enable >2 h acquisitions with no sample evaporation, therefore maintaining a constant salt concentration), and data were acquired by the smfBox. Data were analysed using the MATLAB software package PAM[24]. Bursts were selected using a sliding window dual channel burst search, with a 50 photon threshold and a 500 μs window size. Doubly-labelled bursts were selected between 0.2 and 0.85 S, and bursts were cut into 0.5, 1, and 1.5 ms lengths. To access the precision of the kinetic parameters, acquisitions were split into subsets of 2000 molecules before further analysis. Dynamic PDA[4] was then used to fit a two-state model to the data using the histogram library method implemented in PAM. The raw data for the hairpin experiments (HDF5 files) are available on the smfBox github[18].

**Reporting summary**. Further information on research design is available in the Nature Research Reporting Summary linked to this article.

## Data availability

As keen proponents of open science and open data, we have made the photon arrival time data (photon-HDF5 files) which support the findings of this study publicly available from Zenodo, https://zenodo.org/record/3625603#.XxmB3ChKhm9. This includes data pertaining to: Fig. 2a—Jnotebooks/definitiveset/1ax.hdf5; Fig. 2b, c—Hairpin salt data/1/…; Fig. 2d, e—Hairpin hi-mid-low fret data. The individual rate constant determinations for Fig. 2c are available in the Source Data file. Full build instructions for the smfBox, including animations, are available through our GitHub https://craggslab.github.io/smfBox/[18]. Source data are provided with this paper.

## Code availability

All new acquisition and analysis software are accessible through the Craggs Lab Github (https://craggslab.github.io/smfBox/)[18].

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

## Acknowledgements

The authors thank Gary Turner for refining the design and producing the aluminium pieces for the smfBox, Anna van den Boom for help with AutoCAD and part checking, Lachlan Whitehead for image rendering in Fig. 1b, and Alison Twelvetrees for critical reading and helpful discussions. We are grateful for funding from the following sources: EPSRC studentships (B.A. and E.S.); STFC (B.C.B., M.L.M.-F. and T.D.C); EU Erasmus + (M.A.); The University of Sheffield Start-up grant (T.D.C.); the Royal Society (RGS\R2 \180405, T.D.C), and the BBSRC (BB/T008032/1, T.D.C.).

## Author contributions

T.D.C. designed the research; B.A., J.B., J.C., J.S. and T.D.C. built the instrument; B.A. and M.W. performed all validation experiments and data analysis. B.A., J.B., J.C., E.S. and B.C.B. wrote acquisition and analysis software. J.B. and M.A. performed CAD design and rendering. M.L.M.-F., A.C. and T.D.C. provided supervision. B.A., J.B., J.C. and T.D.C. wrote the manuscript with edits from all authors.

## Competing interests

The authors declare no competing interests.
