## [Peer Review File · Nature Communications]

Reviewers' comments:

Reviewer #1 (Remarks to the Author):

Here, Ambrose et al describe a complete open-source platform for performing single molecule FRET. The hardware construction, acquisition software and analysis software are all provided and thoroughly documented, and the system has been benchmarked using previously-described smFRET standards.

Normally, when reading a paper describing a new microscope build, I tend to admire the design of the system but have little-to-no faith that I could replicate the system myself (for context, I spent my 3-year PhD in an ultrafast spectroscopy lab aligning microscopes and laser paths). Here, however, the description of the hardware construction and set-up to interface with the acquisition computer made me feel that I could definitely build this system myself and I would be confident that it would work as described. This is an incredibly rare trait in any hardware paper. Furthermore, although I obviously could not test the system myself, the smOTTER acquisition software described and screenshotted in the Supplementary Information looks to be of a quality that I would expect from a commercial microscope manufacturer. Having glanced through the Jupyter notebooks for analysis, these are also very well annotated with useful examples.

Having re-read the paper several times, I have been unable to find any substantial flaws (despite trying my utmost, as us reviewers have an annoying habit of doing!). I have some queries that I am happy to be answered either as a reviewer response or perhaps as added discussion to Supplementary Note 9, but would not request any further experiments etc. regarding these points:

- Is there space within the smfBox for temperature control/variation? I think it would be useful/interesting to be able to take smFRET measurements at physiological temperatures.
- Is the sensitivity of the system sufficient for smFRET measurements using e.g. fluorescent proteins? The results shown in the paper are with bright organic fluorophores, but it would also be worth knowing if fluorescent proteins would also work.
- How easy/difficult would it be to exchange the green/red ALEX lasers and associated optics for e.g. blue/green lasers such that other FRET pairs (such as GFP/mCherry) could be investigated?
- For the anodised aluminium base-plate, is this machined to be attached directly to the optical table? If so, are there options for optical tables with metric/imperial holes?

Typos:

- Figure 1d caption: 'unsure' should read 'ensure'
- Supplementary Note 1.2: 'parts A' should read 'part A'

Thank you for an enjoyable paper to read and review!

Signed,

Siân Culley

Reviewer #2 (Remarks to the Author):

Ambrose et al have developed an open-source platform for single-molecule FRET (smFRET). In their manuscript they provide all the necessary details of the hardware and acquisition software such that a newcomer in the field could perform smFRET experiments. They also provide all technical drawings and software including a very good annotation. The platform is very well validated using different dsDNA strands and DNA hairpins. This shows that both high quality distance measurements and kinetic measurements are possible with this platform.

This open-source platform is a very welcome development as smFRET is a technique that is on the rise, but still expensive and difficult for non-specialists.

This manuscript demonstrates that state-of-the-art smFRET data can be obtained using this affordable (homebuilt) setup and open source software. While not providing new insight into biology, I strongly believe that this manuscript is beneficial for a wide audience as reached by Nature Communications.

Some comments:

1. Some arrows in Fig. 1c are confusing. E.g. the arrow from the CCD Camera points towards "Real-time plot...", but in my understanding this is for the "Z Focus Alignment". In turn, the arrow towards the "Z Stage" starts from the "Real-time plot..." , but should probably start from the "Z Focus Alignment".

2. Fig. 1d: The "focal plane" is misleading. For example it looks like the red dye in the left part is out of the focal plane.

3. Fig. 2e: The y-axis has no annotation - it looks like it is the "relative frequency (from Fig. 2d), but it is not.

4. FCS, FCCS, FLCS and PIE have not been demonstrated by this platform. Although I see that FLCS and PIE would be a new project, it seems to be straightforward to show an exemplary FCS and FCCS experiment.

5. In the paragraph "accurate FRET experiments" the authors use a stoichiometry of $S > 0.95$ and $S < 0.175$ while in the paragraph "hairpin dynamics" an S between 0.2 and 0.85 is used. Why are there different numbers and how are they determined?

6. Supplementary Note 7: The authors nicely demonstrate that an asymmetric illumination reduces the width of smFRET histograms. First I wonder if this effect depends on the FRET efficiency (i.e. the distance between the dyes)? Second, the authors state that one reason for the reduction could be a decrease in the shot-noise, but Fig. N7 shows that none of the histograms is shot-noise limited. Third, it seems that the effect depends on the burst selection algorithm. Could the authors expand on that, e.g. by showing this effect for another burst search algorithm? Could the authors advise on how to find the "sweet-spot"?

Reviewer #3 (Remarks to the Author):

The manuscript titled "The smfBox: an open-source platform for single-molecule FRET" by Benjamin Ambrose et al, has provided details including parts and open source software to build a cost-effective diffusion based single molecule FRET spectroscopy. They have measured FRET values of static and dynamic FRET-labeled DNA duplex with a hair pin. Although this manuscript provided high-quality optical design and authentic software, it is highly specialized, on the contrary to their claim of "democratizing for the wider community".

1. Comparing to the more prevalent surface-immobilized, TIRF based smFRET (Nat Methods. 2008

Jun; 5(6): 507–516), the diffusion-based set up has better time resolution. But because the molecules only transiently stay in the focus, long time tracking on one molecule is not possible. In addition, the 50-500s⁻¹ time scale is too fast for many biological processes. Therefore, they need to justify what is the broader application of this setup, in comparison with the TIRF-based smFRET method.

2. To repeat this protocol, significant specialty is required. The TIRF smFRET setup is more friendly for non-specialist. In addition, the new developed CMOS camera lowered the cost significantly. So, there is no advantage of the “low cost” set up described here.

3. The set up described here is more suitable for very fast conformational changes, which is more relevant to biophysicists than to biologists. Economic consideration may not be compatible with gaining meaningful high quality data for very difficult-to-probe processes for physicists. Therefore, it is hard to see which scientific community this publication will find an application.

4. It is hard to see how the dynamic information was extracted in supplement note 8.

In summary, this manuscript has high scientific merit, but it needs to expand extensively on who will be its potential users, what are the suitable processes, and it needs to compare in depth with the TIRF smFRET setup.

We are grateful to all reviewers for their constructive comments and have significantly improved our manuscript in light of their suggestions. In the following point-by-point response, reviewer comments are in blue, our direct responses to these comments are in black, and the changes we have made to the manuscript are indicated in red.

Reviewer 1:

We thank the reviewer for her overwhelmingly positive review. In response to her queries we have made all the requested additions to Supplementary Note 9.

Is there space within the smfBox for temperature control/variation? I think it would be useful/interesting to be able to take smFRET measurements at physiological temperatures.

- The ability to control temperature of the sample would be useful not only for measurements at physiological temperatures, but also to take measurements at a range of temperatures for thermodynamic characterisation of biophysical systems. Given that experiments are usually done on a coverslip alone, an objective heater may suffice to stabilise the sample temperature. We have added a suggested heater to SN9 which should fit between the z control and the top of the box.

We have added the following bullet point to Supplementary Note 9:

- An objective heater (Okolab) to maintain the sample at a physiologically relevant temperature, or a range of temperatures for thermodynamic characterisation.

Is the sensitivity of the system sufficient for smFRET measurements using e.g. fluorescent proteins? The results shown in the paper are with bright organic fluorophores, but it would also be worth knowing if fluorescent proteins would also work.

- Fluorescent proteins have been used on similar systems before (See Orte, Craggs et al JACS 2008, <https://pubs.acs.org/doi/10.1021/ja709973m>).
- We have created an FPbase profile for the smfBox (<https://www.fpbases.org/microscope/Aa5VpxP9ff3nmsdjinYBRUb/>), which can be used to compare both organic fluorophores and fluorescent proteins. A link to this has been added to the smfBox gitsite and in the supplement N1.6 figure legend.
- It seems that there are many green/yellow fluorophores which would show up in the donor channel about as well as our organic fluorophores, in particular mVenus, however there does not seem to be any suitably bright fluorescent proteins in the far-red range which can be excited by the 638 laser. Use of two colour FRET with fluorescent proteins would require a reconfiguration of the lasers and filters, for example a 488/568 combination would likely work well the wealth of green/red FP's (ie. mNeonGreen/mScarlet) which would have comparable brightness to the organic fluorophores typically used in smFRET.

We have added the following section to Supplementary Note 9:

In addition to small organic fluorophores, we expect the smfBox to be capable of detecting fluorescent-protein-labelled biomolecules, as we have shown in the past on similar instruments (Orte, Craggs et al JACS 2008, <https://pubs.acs.org/doi/10.1021/ja709973m>). To this end, we have created an FPbase profile for the smfBox (<https://www.fpbases.org/microscope/Aa5VpxP9ff3nmsdjinYBRUb/>), which allows users to check the compatibility of fluorescent proteins (and other fluorophores) with our instrument, or their version of it (with customisable optics / lasers). However, we note that while there are many green/yellow fluorescent proteins which would be detectable in the donor channel with similar efficiency as our organic fluorophores, e.g. mVenus, there are fewer suitably bright fluorescent proteins in the far-red range which can be excited by the 638 laser. Use of two-colour FRET with fluorescent proteins would therefore require a reconfiguration of the lasers

and filters, for example a 488/568 combination would likely work well the wealth of green/red FP's (ie. mNeonGreen/mScarlet), which we estimate to have comparable brightness to the organic fluorophores we typically use in smFRET.

For the anodised aluminium base-plate, is this machined to be attached directly to the optical table? If so, are there options for optical tables with metric/imperial holes?

- The baseplate diagram supplied is for metric, we will indicate this and show which holes are for the optical table attachment and need to be altered for imperial.

An additional Technical drawing named "Technical_Drawing_smfScope_Box_Base_Plate_HOLES_FOR_NON_METRIC_TABLES" has been supplied.

We have also added the following statement to supplementary note 1:

Note that the smfBox is originally designed for a metric optical breadboard (25 mm spacings). For imperial or any other spacing, the holes in the base plate (A) for attachment to the optical table will have to be machined differently.

Typos: • Figure 1d caption: 'unsure' should read 'ensure' • Supplementary Note 1.2: 'parts A' should read 'part A' Thank you for an enjoyable paper to read and review!

- We have corrected all typos as suggested.

Reviewer 2:

We thank the reviewer for their very positive comments on our manuscript.

1. Some arrows in Fig. 1c are confusing. E.g. the arrow from the CCD Camera points towards "Real-time plot...", but in my understanding this is for the "Z Focus Alignment". In turn, the arrow towards the "Z Stage" starts from the "Real-time plot..." , but should probably start from the "Z Focus Alignment".

We have amended the figure in line with the suggestions – explicitly, we have made sure the arrows discussed point to and away from the Z Focus alignment sub-box.

2. Fig. 1d: The "focal plane" is misleading. For example it looks like the red dye in the left part is out of the focal plane.

We have added "Schematic:" to the legend and the DNA molecule has been shifted slightly so that the red dye is not out of the focal plane.

3. Fig. 2e: The y-axis has no annotation - it looks like it is the "relative frequency (from Fig. 2d), but it is not.

We have added the labels High-FRET, Mid-FRET and Low-FRET to the axis.

4. FCS, FCCS, FLCS and PIE have not been demonstrated by this platform. Although I see that FLCS and PIE would be a new project, it seems to be straightforward to show an exemplary FCS and FCCS experiment.

We agree and now present data in (the new) supplementary note 10. Rhodamine 6G was used as a standard of known diffusion coefficient to characterise the confocal volume, and a

cross correlation of a doubly labelled DNA is shown, which gives a diffusion coefficient of 87 $\mu\text{m}^2/\text{s}$, which compares well with literature values for the 37 bp duplex DNA.

5. In the paragraph "accurate FRET experiments" the authors use a stoichiometry of $S > 0.95$ and $S < 0.175$ while in the paragraph "hairpin dynamics" an S between 0.2 and 0.85 is used. Why are there different numbers and how are they determined?

In the case of accurate FRET experiments high and low S "cuts" are used to isolate the donor and acceptor only populations for determination of correction factors. In the hairpin dynamics no accurate FRET correction was done, so only the mid S doubly labelled population is of interest. These numbers simply correspond to the expected locations of donor only, acceptor only, and doubly labelled species. **We have added a reference to Figure 1f here to aid in understanding.**

6. Supplementary Note 7: The authors nicely demonstrate that an asymmetric illumination reduces the width of smFRET histograms. First I wonder if this effect depends on the FRET efficiency (i.e. the distance between the dyes)?

The width of the histogram does indeed depend on the FRET efficiency, as for any fixed burst size, the histogram takes on the characteristics of a binomial distribution where N is the number of photons per burst and P is the FRET efficiency. (This relationship has been explored in great detail, see Nir et al., 2006). So we would expect a decrease in width at any FRET efficiency, however the decrease would be proportional to the original width at the FRET efficiency.

"Whilst the width in a FRET efficiency histogram is dependent on the mean FRET efficiency, we would expect this effect to still hold at any given FRET efficiency, however the decrease in width would be proportional to the original width at that FRET efficiency."

Second, the authors state that one reason for the reduction could be a decrease in the shot-noise, but Fig. N7 shows that none of the histograms is shot-noise limited.

The noise/width of FRET efficiency histograms is caused by a combination of both shot noise and other sources (background, distance fluctuations etc.). Decreasing shot noise by increasing the number of photons can still reduce the total noise even if shot noise is not the only kind of noise present. We have reworded this section to reiterate that shot noise is not the only contributing noise but does represent a considerable proportion of it.

...as shot noise is not the only form of noise present, but represents a considerable proportion, nonetheless. Other forms of noise which make up for the discrepancy are background contribution, and flexing of the DNA molecule, which has been previously discussed in detail^{4,24,25}.

Third, it seems that the effect depends on the burst selection algorithm. Could the authors expand on that, e.g. by showing this effect for another burst search algorithm? Could the authors advise on how to find the "sweet-spot"?

The effect likely would depend on the parameters used in the burst selection algorithm as the amount of time spent illuminating with either laser directly affects the number of photons detected in that channel and hence the likelihood that any given burst would meet the threshold applied by the algorithm. A proper investigation into finding the "sweet-spot" would require an exploration of multiple inter-dependent parameters of laser power, burst selection thresholds, photo-bleaching, etc. We felt such an investigation was beyond the scope of the current paper, which focuses on presenting a new instrument and features which can be controlled by the open-source acquisition software. This is why we chose to demonstrate a possible advantage of an asymmetric excitation scheme with all other things being equal, to

establish the value of the smfBox having an easily variable duty cycle, rather than as an in-depth characterisation of asymmetric ALEX. Furthermore, the exact sweet spot may vary depending on the precise application, as there may be circumstances where more bursts are desired (quantifying relative amounts of high/low FRET species), and some where less histogram width is desired (single species accurate FRET determination), so a full optimisation is not possible without knowledge of the intended purpose.

We have clarified this further in the main text:

From this plot we would recommend **as an immediate improvement over symmetric ALEX** a 'sweet-spot' of 60G-30R for the duty cycle, to both maximise the number of detected bursts and minimise the shot noise. **The exact sweet spot may vary depending on the application however; if trying to detect a relatively small shift in FRET efficiency then tighter histograms at the expense of fewer bursts may be desirable, however if quantifying two populations which are already well separated in FRET efficiency, then tighter histograms may be less of an advantage. With a large parameter space (laser power, photobleaching, oxygen scavengers, burst length, burst size, noise, etc.) it is difficult to prescribe an objectively optimum aALEX ratio for any given application. However, the ability to alter the duty cycle available in the software provided allows the user to optimise as they see fit.**

Reviewer #3:

We thank the reviewer for their comments and address them below.

1. Comparing to the more prevail surface-immobilized, TIRF based smFRET (Nat Methods. 2008 Jun; 5(6): 507–516), the diffusion-based set up has better time resolution. But because the molecules only transiently stay in the focus, long time tracking on one molecule is not possible. In addition, the 50-500s-1 time scale is too fast for many biological processes. Therefore, they need to justify what is the broader application of this setup, in comparison with the TIRF-based smFRET method.

We agree that our manuscript did not explicitly discuss the different strengths of confocal and TRIF smFRET methods. To address this, we have added two paragraphs to the introduction (shown below), which explain these, and also the key biological processes which can be studied using the confocal method.

Two experimental formats are commonly employed to obtain smFRET data: a confocal approach, in which individual molecules are detected as they diffuse through a confocal volume³; and a total internal reflection fluorescence (TIRF) microscopy approach¹², in which individual molecules are immobilised on a glass coverslip and excited by an evanescent wave. The two approaches are largely complementary, together allowing the interrogation of biomolecular dynamics at timescales spanning twelve orders of magnitude: from the picosecond – millisecond (confocal); and millisecond – hours (TIRF) (see ref ¹³ for a recent review). Confocal experiments are often the first smFRET experiments performed on a new biomolecular system, as this is the more straightforward approach, not requiring surface immobilisation (which can be non-trivial). Such experiments can reveal different biomolecular conformations present in (dynamic) equilibrium, and the corresponding rates of conformational transitions, at timescales relevant to key cellular processes such as protein (un)folding¹⁴, transcription¹⁵ and DNA replication and repair^{16,17}. Confocal experiments are also the approach of choice for generating multiple smFRET restraints for integrative structural modelling⁶⁻¹¹, due to the simpler sample preparation, fast data acquisition, and higher time resolution.

Despite the **many** advantages of smFRET, it is currently rarely used outside specialist labs, largely due to the high costs of commercial instruments and lack of self-build, easy to use alternatives. **To address this, here we provide detailed build instructions, parts lists, and open-source acquisition software, to enable a broad range of scientists to perform confocal smFRET experiments, on a validated, self-built, robust and economic instrument.**

2. To repeat this protocol, significant specialty is required. The TIRF smFRET setup is more friendly for non-specialist. In addition, the new developed CMOS camera lowered the cost significantly. So, there is no advantage of the “low cost” set up described here.

Whilst we partially agree with the reviewer that some specialist knowledge is required for the implementation of smFRET on the smfBox, we do not agree that TIRF smFRET experiments are ‘more friendly for non-specialist’. As we discussed in our response to point 1 (above, and in additions to the main text), TIRF approaches require surface immobilization of the biomolecules of interest, and extensive surface passivation. These steps are non-trivial and require considerable optimisation before good quality data can be obtained. Confocal experiments require no such slide preparation, and are conducted by pipetting the sample onto a simple coverslip, making the data acquisition much more straightforward. Other aspects of the experiment require similar levels of skill.

We agree that camera technology is improving all the time, and cheaper, bigger CMOS chips are a welcome innovation over the expensive EM-CCD chips traditionally used for smFRET TIRF. However, as noted by the reviewer above, the time resolution achieved by current camera-based detection (down to around 1 millisecond) is much slower compared to the nanoseconds achieved by the avalanche photodiodes used in confocal approaches. Both approaches are useful and give access to different timescales, as we note in the paragraphs we have added to the introduction. We do not see this instrument as a competitor to TIRF, rather than as a complement.

3. The set up described here is more suitable for very fast conformational changes, which is more relevant to biophysicists than to biologists. Economic consideration may not be compatible with gaining meaningful high quality data for very difficult-to-probe processes for physicists. Therefore, it is hard to see which scientific community this publication will find an application.

We agree that biophysicists are the likely early adopters of our new instrument. We would argue that reducing the entry barrier in terms of cost and ease of use, can only increase the uptake of this method by the community. There are numerous biophysics labs which are keen to do these types of experiments but can not justify buying a commercial instrument at ten times the price of assembling the smfBox. We also anticipate that in the mid-term, a wider range of labs (e.g. structural biologists, enzymologists, biochemists etc) will adopt this technology, especially given its ease of use, and powerful capabilities.

4. It is hard to see how the dynamic information was extracted in supplement note 8.

We have added additional detail (below) to Supplementary Note 8 to clarify how FRET-2CDE is used, however the specifics of how FRET-2CDE works are explained in detail in the cited reference (28).

Both methods use the photon arrival times within each burst to “score” for dynamics, BVA uses the variance of FRET efficiencies of subdivided bins within each burst, whereas 2CDE compares kernel density estimators between channels. As each burst is scored for discrimination between static/dynamic, these methods can be used as hypothesis testing to verify that dynamic populations are present before being quantified by dPDA.

REVIEWERS' COMMENTS:

Reviewer #1 (Remarks to the Author):

The authors have satisfactorily addressed my comments from the previous version of the manuscript, thank you!

My only remaining very minor comments (some of which are things that I probably missed last time round, which I apologise for) are:

- In the introduction (lines 23-27), the authors switch usage between 'chromophore' and 'fluorophore'. This may be intentional (I trust the authors' expertise on this more than my own!) but I wanted to flag it just in case
- Typo on line 63 – "adonised" should read "anodised"
- Supplementary Information line 26 – I think "Supplementary Fig. N.2" should instead refer to N1.2
- Typo on line 453 of Supplementary Information – "aALEX" should read "ALEX".

Signed,
Siân Culley

Reviewer #2 (Remarks to the Author):

The authors have very well addressed all my comments. In particular they show now convincing FCS data of Rhodamine 6G and a doubly labelled DNA in the new Supplementary Note 10. In my view this will largely increase the applicability of this instrument.

Just the variables in Supplementary Table N10.1 should be defined (although the variables largely speak for themselves).

I fully support publication of this revised manuscript.

Response to referees

We thank the reviewers for their time and constructive comments. We have made all the suggested changes.

Reviewer 1

- In the introduction (lines 23-27), the authors switch usage between 'chromophore' and 'fluorophore'. This may be intentional (I trust the authors' expertise on this more than my own!) but I wanted to flag it just in case.

It was intentional and we have left the text as it is. For reference, the donor needs to be a fluorophore (ie emits fluorescence), the acceptor can be a chromophore (ie absorbs and is therefore coloured, but does not necessarily emit). A fluorophore is by definition a chromophore, but a chromophore is not necessarily a fluorophore.

- Typo on line 63 – “adonised” should read “anodised”

Corrected (good catch!).

- Supplementary Information line 26 – I think “Supplementary Fig. N.2” should instead refer to N1.2

Corrected.

- Typo on line 453 of Supplementary Information – “aALEX” should read “ALEX”.

aALEX is short for asymmetric alternating laser excitation, but to make this easier to understand we have spelt this out.

Reviewer 2

The authors have very well addressed all my comments. In particular they show now convincing FCS data of Rhodamine 6G and a doubly labelled DNA in the new Supplementary Note 10. In my view this will largely increase the applicability of this instrument.

Just the variables in Supplementary Table N10.1 should be defined (although the variables largely speak for themselves).

I fully support publication of this revised manuscript.

We have added foot notes to fully define the variables in Table N10.1